# Green Synthesis of Transition-Metal Nanoparticles and Their Oxides: A Review

**DOI:** 10.3390/ma14112700

**Published:** 2021-05-21

**Authors:** Sean Drummer, Tafirenyika Madzimbamuto, Mahabubur Chowdhury

**Affiliations:** Department of Chemical Engineering, Cape Peninsula University of Technology, Symphony Way, Bellville 7535, South Africa; sndrummer512@gmail.com (S.D.); MadzimbamutoT@cput.ac.za (T.M.)

**Keywords:** nanoparticle, transition-metal, transition-metal oxides, plants, green synthesis, factors

## Abstract

In recent years, many researchers have begun to shift their focus onto the synthesis of nanomaterials as this field possesses an immense potential that may provide incredible technological advances in the near future. The downside of conventional synthesis techniques, such as co-precipitation, sol-gel and hydrothermal methods, is that they necessitate toxic chemicals, produce harmful by-products and require a considerable amount of energy; therefore, more sustainable fabrication routes are sought-after. Biological molecules have been previously utilized as precursors for nanoparticle synthesis, thus eliminating the negative factors involved in traditional methods. In addition, transition-metal nanoparticles possess a broad scope of applications due to their multiple oxidation states and large surface areas, thereby allowing for a higher reactivity when compared to their bulk counterpart and rendering them an interesting research topic. However, this field is still relatively unknown and unpredictable as the biosynthesis of these nanostructures from fungi, bacteria and plants yield undesired diameters and morphologies, rendering them redundant compared to their chemically synthesized counterparts. Therefore, this review aims to obtain a better understanding on the plant-mediated synthesis process of the major transition-metal and transition-metal oxide nanoparticles, and how process parameters—concentration, temperature, contact time, pH level, and calcination temperature affect their unique properties such as particle size, morphologies, and crystallinity.

## 1. Introduction

Due to recent advancements and developments in industry, the environment has suffered tremendously with toxic wastes regularly being released into the ecosystems, causing irreparable damage to biological life and infrastructures. Therefore, it is necessary for all engineering and technological aspects to move towards a more sustainable outlook. One of the most rapidly developing industries is the “nano industry,” where nanotechnology could possibly bring about a new industrial revolution.

Nanotechnology refers to the field of applied science and technology, which involves the manipulation of matter on a molecular and atomic scale with several prospective applications [1]. Nanoparticles are defined as specific complexes of solid units that possess sizes between 1 and 100 nm. The most unusual aspect of nanoparticles is that they retain a large surface area due to the minute size of the particles when compared to their bulk counterpart. This characteristic of the nanomaterial allows for higher reactivity and denotes specialized properties to them [2]. Due to these attributes, nanoparticles may be applied in chemical, biological, and industrial fields.

Transition metals are defined as *d*-block elements as their atomic structures have incomplete *d*-orbitals in their electron configuration. Therefore, they give rise to physical and chemical properties that differ from the main group elements [3]. These elements are considered interesting for research as they retain multiple oxidation states caused by the proximity in the energy of the *d* and *ns* shells [4]. This provides a large number of configurations for their oxides, thus widening the field of applications for transition metals.

Specific physical and chemical processes have been established for the synthesis of metal nanoparticles and their oxides. Through copious research and experimentation, these methods have become well-defined, where the physicochemical properties of the nanostructures can be excellently controlled and tailored as needed. The conventional techniques have been listed under top-down and bottom-up methods in Figure 1. However, these procedures have several disadvantages as they require the use of toxic chemicals, produce harmful by-products and necessitate a high energy consumption [5]. In addition, these processes are quite troublesome to scale up if industrial-scale manufacturing is required. Hence, it is essential for the further development of the environmentally friendly approach to synthesize metal and metal oxide nanomaterials.

Through previous studies, it has been established that various biological routes may be utilized for the fabrication of nanoparticles through the use of plants [6], bacteria [7], fungi [8], algae [9], and yeast [10], as they contain metabolites that possess the ability to reduce metallic salts and formulate nanoparticles. In addition, these substances do not only act as a reducing agent as they are simultaneously involved in the stabilization of the nanostructures. Since this area of “green” nanotechnology is relatively new [11], the problem lies with controlling the size and morphology of the nanoparticles due to the different characteristics and compositions of the biological mediums utilized during synthesis. This being said, the main objective is to achieve a better understanding of the green synthesis to acquire the desired size, shape, composition, and dispersity, with an exceptional degree of repeatability if conventional approaches are to be replaced.

Recent studies have been successful with innovative nanoparticle applications such as cancer treatment [12], antibacterial activity [13], drug delivery systems [14], antioxidant activity [15], catalytic activity [16], and the removal of toxic pollutants from wastewater [17]. Compared to the chemically synthesized counterparts, biologically produced nanoparticles have proven superior due to the organic capping molecules and defects in their crystalline structure, ultimately enhancing their reactivity [18,19].

Another advantage of the biological synthesis route is the possibility of industrial-scale manufacturing. To date, there is no known operation for the industrial fabrication of nanoparticles through these techniques. However, this would be beneficial as the primary raw material is renewable, and the process would require minimal energy, thereby resulting in low operating costs and almost negligible toxicity in waste. Therefore, the entire system would be completely sustainable.

Herein, this review will focus on summarizing the current progress in the plant-mediated synthesis of various transition-metal nanoparticles such as titanium, iron, cobalt, nickel, copper, zinc, palladium, silver, platinum, gold, and their oxides. In addition, the advantages over the conventional synthesis techniques, green synthesis reaction mechanism, possible compounds responsible for the reduction reaction and critical process parameters are discussed. Ultimately, the objective is to present the procedures and results of green synthesized transition-metal and metal oxide nanoparticles for the enhancement of the literature in this field which may aid researchers in their future endeavours.

## 2. Biological Synthesis Techniques

Within the past three decades, solutions to the disadvantages of conventional nanoparticle techniques have been sought after. The main goal is to reduce the negative aspects of the aforementioned chemical methods (Figure 1) and create more sustainable methods to minimize pollution at the source instead of necessitating waste management procedures. Interestingly, nature provided the tools for a new revolutionary technique that utilizes plants and microorganisms for nanoparticle fabrication.

It was discovered that most organisms that produce their own nutritional compounds using organic material in their vicinities, such as plants, fungi, bacteria, and yeast, possess a particular reducing capability that enables the material to react with metallic ions and form any desired nanoparticle [20,21,22,23]. The “green” synthesis of nanoparticles has some advantages over the conventional methods as they limit the use and production of toxic, inorganic chemicals and can be carried out in ambient conditions while still preserving the quality of the nanostructures with a relatively fast production rate [24].

The procedure involves metal ions that undergo either oxidation or reduction, depending on the biological media utilized. The exact mechanisms for the biological synthesis of nanoparticles have not yet been extensively studied, as each autotroph contains different compounds responsible for the reaction. However, it is known that the process is initiated through the nucleation of the metallic substance and, thereafter, nanoparticle growth commences where it is formed as the precipitate within the mixture (Figure 2). Furthermore, the organic material found in plants and microorganisms is also involved in the stabilization, in other words, the capping of the nanoparticles as they bind to the surface of the nanostructures, thereby preventing aggregation and negating the need for additional stabilizing agents [25].

According to research, plant extracts seem to be the superior biological mediums as they are more stable, reduce metal ions faster, and are easier to scale up when compared to microorganisms [25,26,27]. Therefore, plants have been the main focus of research in the biosynthesis of nanoparticles as they also retain a high synthesis success rate in contrast to other “green” methods [28].

Previous studies have shown that the primary and secondary metabolites of plants are responsible for reducing metallic ions during the synthesis procedure [5]. These major phytochemicals are categorized within the Appendix A, in Appendix A. They include terpenoids, phenols, carbohydrates, saponins, alkaloids, and proteins. These compounds are expected to induce some shape control during the reduction reaction [29]. Hence, the variation in composition and concentration of the active biomolecules between plant species and their behaviour with the metallic reactants is believed to be the main contributing factors to the diverse aspects of the nanomaterial, and the screening of phytochemicals with higher electron-donating capacities may positively impact future research in the plant-based synthesis of nanoparticles [28,30]. This being said, the optical and electrical properties of the nanoparticles are highly dependent on the size and morphology, and it is essential to achieve complete control of these characteristics for further development in the field of “green” nanotechnology.

## 3. Green Synthesis of Transitional Metals and Their Oxides

The synthesis of nanoparticles using the aqueous extracts of plant materials is a simple procedure that merely involves the use of plant extract as a reducing agent and a metallic salt solution (Figure 3). Transition-metal nanoparticles have been the prime focus of researchers due to the vast and interesting applications they possess. Section 3 focuses on the green synthesis of titanium-, iron-, cobalt-, nickel-, copper-, zinc-, palladium-, silver-, platinum-, and gold-based nanoparticles, from 2005 to date, which includes various process parameters with their resulting particle sizes and morphologies. This was accomplished by analyzing published articles that involve various plant extracts for the synthesis of transition-metal and transition-metal oxide nanoparticles.

### 3.1. Titanium and Titanium Dioxide

The biosynthesis of pure titanium nanoparticles is extremely rare; however, one of the only attempts to synthesize the nanostructures was executed through the use of *Azadirachta indica* in a study conducted by Krishnasamy et al. (2015). After characterization, the nanomaterial was crystalline in nature with a mixture of rutile and anatase phases. In addition, the nanostructures possessed spherical morphologies with a size range of 15 to 42 nm [31].

Kalyanasundaram and Prakash (2015) demonstrated the green synthesis of titanium dioxide nanoparticles using *Pithecellobium dulce* and *Lagenaria siceraria* aqueous leaf extracts from titanium tetra-isopropoxide (Ti(OCH(CH_3_)_2_)_4_) as a precursor. Their study compared the biologically synthesized nanocrystals to those of the chemical synthesis methods, where the sol-gel technique was adopted as the chemical synthesis route. The nanoparticles synthesized were predominantly spherical, with the *Pithecellobium dulce* synthesis obtaining the smallest particle size of approximately 6.3 nm and the chemically synthesized titanium dioxide possessing the largest diameter of 14.4 nm [32]. Furthermore, a study conducted by Dobrucka (2017) resulted in the successful synthesis of titanium dioxide nanoparticles with the use of aqueous *Arnicae anthodium* extract as a bioreducing and capping agent [33]. The XRD study conducted on the nanopowder confirmed the average particle size to be approximately 30 nm.

Previous studies that focused on the plant-mediated synthesis of titanium dioxide nanoparticles (Appendix A) produced mainly spherical particles that possessed a particle size of approximately 40 nm. However, there were occasions where unusual morphologies were experienced, such as tetragonal, oval, rod, and triangular shapes which were synthesized through the use of *Citrus sinensis*, *Vigna radiata*, *Taraxacum officinale*, and *Sesbania grandiflora* extracts [34,35,36,37]. In addition to this, the titanium dioxide fabricated with *Euphorbia prostrata* and *Hibiscus rosa-sinensis* experienced a degree of polydispersity throughout the nanostructures as an undesired result from the synthesis process [38,39].

### 3.2. Iron and Iron Oxide Nanoparticles

A study conducted by Hoag et al. (2009) was one of the first recorded articles for the green synthesis of zero-valent iron nanoparticles. Their research utilized the polyphenols in *Camellia sinensis* to act as the bioreducing agent for different ferric precursors such as FeCl_3_, FeSO_4_, and Fe-EDTA at room temperature [40]. It was found that the zero-valent iron nanocolloids were spherical in nature and possessed diameters ranging from 5 to 15 nm. These zero-valent iron nanoparticles retained similar properties to iron nanoparticles synthesized through chemical reduction, where the nanocrystallites were discovered to be highly monodispersed with diameters of 6 nm [41].

Latha and Gowri (2014) published their findings on the synthesis of magnetite (Fe_3_O_4_) using the aqueous extract of *Caricaya papaya* leaves. The SEM images of the nanomaterial indicated the morphology plate-like structures with coarsened grains and capsules, whilst the XRD analysis found that the average particle size of the magnetite was 33 nm (Figure 4).

Furthermore, Pravallika et al. (2019) recently studied the green synthesis of iron oxide nanoparticles using the extract of *Centella asiatica* by reducing ferrous and ferric chlorides. The TEM image of the nanopowder displayed agglomerated spherical particles ranging from 20 to 40 nm in size. According to the FTIR analysis, it was confirmed that the nanostructures were capped with biomolecular compounds such as triterpenoids [43]. 

In addition to the above, iron nanoparticles were synthesized using *Lawsonia inermis* and *Gardenia jasminoides* extract in a study conducted by Naseem and Farrukh (2015). The TEM results obtained from the *Lawsonia inermis* extract displayed a particle size of 21 nm, while the latter’s iron nanoparticles were observed to be 32 nm. The SEM images of the extracts exhibited clustered nanostructures with hexagonal and shattered rock-like appearances for *Lawsonia inermis* and *Gardenia jasminoides* extracts, respectively [44].

Moreover, Yuvakkumar et al. (2014) synthesized magnetite (Fe_3_O_4_) nanoparticles using *Nephelium lappaceum* waste extract as a green ligation and chelating agent. The XRD study disclosed the iron oxide nanoparticles to possess a spinel structure, while the TEM images revealed an average particle size of 200 nm. 

The synthesis of iron nanoparticles and their oxides was strongly influenced by the concentration of antioxidant compounds within the various plant extracts used, where the plants with high phenolic content were found to display the greatest reductive capabilities [45,46]. Despite some anomalies, most iron-based nanoparticles were discovered to be spherical with slight agglomeration and an average particle size of approximately 58 nm (Appendix A) [47,48].

### 3.3. Cobalt and Cobalt Oxide Nanoparticles

In 2015, a study conducted by Sharma and co-authors displayed the successful synthesis of Co_3_O_4_ nanoparticles using the aqueous extract of *Calotropis gigantea* leaves. From the SEM and TEM results, nearly spherical shaped particles were observed with diameters ranging from approximately 60–80 nm. The energy dispersive X-ray (EDX) spectrum found Co and O to be the significant elements, indicating the cobalt oxide nanoparticles to be of high purity [49].

Interestingly, Han and co-authors (2015) established the green synthesis of three-dimensional hierarchical porous cobalt oxide nanostructures using mature *Gingko biloba* leaves as a growth template. After rinsing and drying, the plant material was introduced into a 0.1 M cobalt(II) acetate (Co(CH_3_COO)_2_) solution where the mixture was left at room temperature overnight. The proposed reaction for the reduction in the cobalt salt is displayed in Equation (1). The TEM and SEM images deduced that the nanoparticles possessed a hierarchical, porous worm-hole-like structure with sizes ranging from 30–100 nm [50].
(1)$$3\mathrm{Co}{\left({\mathrm{CH}}_{3}\mathrm{COOH}\right)}_{3}+8O_{2}\to {\mathrm{Co}}_{3}O_{4}+12{\mathrm{CO}}_{2}$$

Furthermore, cobalt nanoparticles were synthesized using the biomolecules extracted from *Conocarpus erectus* [51]. The study utilized methanol, ethanol, and deionized water as different extraction solvents, resulting in the methanolic extract bearing the highest concentration of phenolic compounds (296 µg/g) and, therefore, most significant antioxidant capacity. The SEM results disclosed cobalt nanoparticles possessing spherical morphologies and diameters ranging from 20 to 60 nm. On the other hand, cobalt nanoparticles produced through the liquid-phase reduction method possessed spherical shapes with larger diameters of 400 nm, ultimately decreasing the reactivity compared to the biologically synthesized Co-NPs [52].

Other interesting morphologies such as octahedral, hexagonal, pentagonal, plate-like, and spinel were obtained through *Manihot esculenta crantz*, *Chromolaena odorata*, *Mangifera indica*, *Helianthus annus*, and *Moringa oleifera*, respectively (Appendix A) [53,54,55,56,57]. Additionally, the average nanoparticle size of the cobalt-based materials was found to be approximately 54 nm, with multiple studies experiencing agglomeration [58,59]. Moreover, Diallo and co-authors (2015) discovered that phenolic compounds exhibited high antioxidative properties and were the major contributors to the reduction in metal ions. In contrast, plants with high concentrations of proteins, lipids, and amino acids contribute to stabilizing the nanostructures and, in turn, inhibits the formation of agglomerates [60].

### 3.4. Nickel and Nickel Oxide Nanoparticles

One of the first recorded studies that utilized plant extract for the synthesis of nickel nanoparticles (Ni-NPs) was conducted by Sudhasree and co-authors (2014). Their research employed the aqueous extract of *Desmodium gangeticum* root, where the extraction process was conducted using the Soxhlet method. The ultraviolet-visible (UV-vis) spectrophotometer confirmed the synthesis of spherical Ni-NPs due to a sharp band at 270 nm [61]. Another study employed *Camellia sinensis* as a reducing and capping agent for the synthesis of nickel nanostructures, resulting in spherical morphologies and an average particle size of 42 nm [62].

Nickle oxide nanocrystallites were synthesized via the green route using *Aloe vera* by Juibari and Eslami (2019). The SEM results displayed the morphology of nanorods with homogenous structures with an average diameter of 50–70 nm [63]. Furthermore, NiO-NPs were prepared using *Physalis angulata* leaf extract in recent experimentation conducted by Sulaiman and Yulizar (2018). The electron microscope studies displayed the morphology of spherical particles, whilst the particle size analysis found a homogenous size distribution with a sharp peak at 64.13 nm [64]. Nickel oxide nanostructures produced through chemical precipitation [65] and sol-gel [66] techniques resulted in uniform sizes of 35–45 and 32.9 nm, respectively (Figure 5). The nanomaterials possessed spherical morphologies and were formed without impurities, deeming them slightly superior to the plant-mediated NiO-NPs.

In summary, most studies obtained spherical morphologies and face centred cubic structures, with an average particle diameter of 34 nm (Appendix A). Once again, aggregation of the nanostructures was an issue for researchers during the synthesis of these nickel-based compounds. This may have been caused by the presence of cell components on the surface of the nanocolloids or the magnetic interaction and polymer adherence between the particles [67,68,69,70,71]. Additionally, a study conducted by Chen and co-authors (2014) mentioned that the flavonoids and sugars of alfalfa extract were the main reducing agents for Ni(II), confirming the involvement of these phytochemicals during the reduction process [72].

### 3.5. Copper and Copper Oxide Nanoparticles

Nasrollahzadeh and co-authors (2014) reported synthesizing copper nanoparticles using the aqueous extract of the *Euphorbia esula* leaves. The morphology of the nanomaterial was found to be spherical, with sizes distributed between 20 and 100 nm, according to the TEM images [73]. Similarly, *Phyllanthus embilica* plant extract was previously used to synthesize copper nanostructures [74]. The SEM results exhibited the shape of the nanoparticles to be flake-like with diameters ranging from 15 to 30 nm.

Copper oxide nanoparticles were synthesized using chemical and biogenic methods by Muthuvel and co-authors (2020). The study utilized *Solanum nigrum* extract as the green synthesis route and the sol-gel technique as the latter. The High-Resolution TEM (HR-TEM) analysis of the nanomaterials revealed spherical particles with diameters of 32 and 25 nm for chemical and biological synthesis, respectively. The authors stated that the biosynthesized nanoparticles had slight structural differences compared to the chemically produced colloids due to the thin organic coating of the particle. The nanoparticles also showed better dispersion within the solution and higher porosity, enhancing reactivity towards applications [75].

Moreover, the green synthesis of copper oxide nanostructures was investigated using *Ocimum basilicum* extract by Altikatoglu and co-authors (2017). FTIR spectroscopy was used to identify the functional groups based on the bands from the scan. The peak at 3393 cm^−1^ was attributed to the O-H stretching vibrations of alcohols and phenols. The bands situated at 1600 and 510 cm^−1^ formed due to the stretching of Cu-O, which confirmed the synthesis of copper oxide nanoparticles [76].

In a recent study, Jayarambabu and co-authors (2020) successfully synthesized copper nanostructures using the aqueous extract of *Curcuma longa*. The ethanolic plant extract was mixed with 0.1 M copper acetate dihydrate and introduced into a microwave for 180 s at 200 W power. The HR-TEM and Field Emission SEM (FESEM) images displayed particles with spherical morphologies and diameters ranging from 5 to 20 nm [77].

The copper and copper oxide nanoparticles were found to possess spherical shapes with an average size of 53 nm (Appendix A). Numerous researchers conducted FTIR characterization techniques on the extract solutions and nanostructures which found that the reduction in the metallic ions resulted from phytochemicals such as phenols, flavones, terpenoids, proteins, and other heterocyclic compounds present during the reaction [78,79,80,81]. Copper-based nanomaterial studies also indicated initial aggregation of the particles that increased over time [82,83]. However, it was denoted that agglomeration may have been caused by an increase in catalytic activity on the surface of the nanoparticles, leading to the enhancement of its application [84].

### 3.6. Zinc and Zinc Oxide Nanoparticles

A study conducted by Sangeetha and co-authors (2011) was one of the first reported green synthesis of zinc oxide using plant extract. The research utilized *Aloe barbadensis miller* leaf extract as a reducing and capping agent for the selected precursor. The SEM and TEM monographs displayed clustered ZnO nanoparticles with spherical and hexagonal morphologies and particle sizes ranging from 25 to 55 nm [85].

Furthermore, zinc oxide nanoparticles were formulated using the aqueous extract of *Lantana aculeata* leaf [86]. The UV-vis spectrum of the nanomaterial recorded a peak at 362 nm, confirming the presence of ZnO nanoparticles. The FTIR studies further confirmed the successful green synthesis of the metal oxide nanomaterial with the peak at 443.63 cm^−1^ allotted to ZnO, caused by the zinc and oxygen bond stretching. The remaining peaks at 1028.06, 1097.5, 1757.15, and 3481.51 cm^−1^ were due to the presence of the compounds amongst the nanostructures. These include aliphatic amines, carboxylic acids, alcohols, and phenols. The morphological analysis carried out on the substance displayed FESEM and HR-TEM images with spherical ZnO particles and an average size of 12 ± 3 nm. Madathil and co-authors (2007) hydrothermally synthesized zinc oxide nanostructures similar to the *Lantana aculeata*-mediated colloids [87]. The particle diameters were found to be about 7–24 nm, with both studies obtaining highly crystalline structures and XRD peaks corresponding with the standard wurtzite structure. 

The majority of the studies utilized the green synthesis method to fabricate zinc oxide nanoparticles rather than pure zinc nanostructures. However, zinc nanoparticles were successfully synthesized through the use of *Andrographis paniculata*, *Cestrum nocturnum*, *Solanum nigrum*, and *Spilanthes acmella* extracts which produced spherical and wire-like morphologies with particle sizes ranging from 10 to 70 nm [88,89,90,91]. The average particle size of zinc-based nanostructures was found to be approximately 44 nm and possessed spherical morphologies with hexagonal structures (Appendix A). Other interesting morphologies such as floral, cylindrical, and sponge-like shapes were obtained using *Phyllanthus emblica*, *Parthenium hysterophorus*, and *Hibiscus rosa-sinensis*, respectively [92,93,94]. Despite the high purity of the zinc and zinc oxide nanostructures, many researchers discovered their resulting nanoparticles to be agglomerated due to electrostatic attractions [95,96,97].

### 3.7. Palladium and Palladium Oxide Nanoparticles

In 2009, Sathishkumar and co-authors reported one of the first studies that utilized *Cinnamom zeylanicum* extract for the synthesis of palladium nanoparticles (Pd-NPs). The TEM and XRD analysis confirmed the successful production of palladium nanoparticles with spherical morphologies and diameters ranging from 15 to 20 nm [98]. In addition, *Moringa oleifera* extract was used in the microwave-assisted green synthesis of Pd-NPs [99]. A mixture of 1 mM palladium acetate (Pd(OAc)_2_) and the methanolic plant extract was irradiated at 300 W for 5 min. The morphological studies of the nanoparticles revealed agglomerated particles with spherical shapes and an average size of 27 nm (Figure 6).

Moreover, Sharmila et al. (2017) conducted experiments that successfully synthesized palladium nanoparticles by utilizing the aqueous leaf extract of *Filicium decipiens*. The TEM images displayed spherical shapes with the size distribution of 2–22 nm and a mean particle size of 6.36 nm [100]. Palladium nanocolloids prepared through the polyol method, and stabilized with polyvinylpyrrolidone (PVP), displayed profound shape and size control. High concentrations of AgNO_3_, FeCl_3_, and NaI were used as controlling agents, resulting in morphologies of spheres, cubes, tetrahedrons, octahedrons, decahedrons, and icosahedrons ranging from 2 to 30 nm [101]. The plant-mediated synthesis of PdO-NPs has not been as extensively explored as palladium nanomaterials; however, it was found that *Aspalathus linearis* extract successfully produced pure tetragonal crystallites that possessed spherical particles and diameters ranging from 3.8 to 22 nm [102]. 

Previous research on the biosynthesis of plant-conjugated palladium nanoparticles obtained primarily spherical particles with a mean size of approximately 24 nm, which is relatively small compared to other metallic nanostructures (Appendix A). Additionally, detailed analysis of the plant extracts used for the biosynthesis of palladium nanoparticles indicated that flavonoids, polyphenols, saponins, tannins, anthocyanins, betacyanins, terpenoids, steroids, and proteins served as the critical donors of electrons during the redox reaction for the fabrication of Pd nanoparticles. These phytochemicals are absorbed onto the surface of the nanostructures through *π*-electron interactions if other strong ligating agents are not present [100,103,104]. Once again, agglomeration was an issue faced by researchers, deeming the particles thermodynamically unstable as they are, by nature, finely divided mass [105,106].

### 3.8. Silver and Silver Oxide Nanoparticles

In a study conducted by Gopinath and co-authors (2012), silver nanoparticles were biosynthesized using the extract of *Tribulus terrestris*. The TEM and Atomic Force Microscope (AFM) analysis methods determined the size and morphologies of the nanomaterial, resulting in average particle sizes of 22 nm and spherical shapes [107]. The green synthesis route of silver nanoparticles was enhanced through the use of photo-irradiation and *Osmanthus fragrans* extract. The experimentation involved the reaction to take place in bright sunlight, where silver nanoparticles were formed within 2–3 min as a colour change was observed. The microscopy studies revealed spherical nanostructures with particle sizes of 20 ± 3 nm [108]. Similarly, solar irradiation assisted the synthesis of *Nageia nagi*-mediated silver nanocolloids [109]. For this study, the Ag-NPs were photo-reduced within 25 s after exposure to sunlight, deeming this technique exponentially faster than regular green synthesis routes. Morphological studies concluded that the particles were spherical with an average diameter of 16.1 nm.

Furthermore, medicinal plants such as bamboo leaves have successfully been used to produce silver nanoparticles. These plants possess antiviral and antibacterial qualities as they have an abundance of phytochemicals such as flavonoids, phenolic acids, and lactones [110]. Yasin and co-authors (2013) found that their silver nanoparticles were nearly spherical, with diameters of 100 nm and smaller. In addition, the nanomaterial exhibited suitable antibacterial activities against various samples and could potentially be utilized for different medical applications.

Despite the aforementioned results, silver nanoparticles prepared through co-precipitation [111] and sol-gel [112] techniques achieved pure crystalline structures with regular granular shaped grains and diameters of 5.5 and 11 nm, respectively. Therefore, resulting in slightly greater reactivity due to an increase in surface area compared to the plant-mediated nanocolloids.

Although not as extensively studied as pure silver nanoparticles, silver oxide nanoparticles have been successfully synthesized through the use of plant extracts [113,114]. More specifically, Ravichandran and co-authors (2016) fabricated Ag_2_O nanostructures through *Callistemon lanceolatus* extract as a reducing and capping agent. The redox reaction (Equations (2) and (3)) was initiated by mixing 5 mL of the aqueous extract with 100 mL of 1 mM AgNO_3_ and incubated at 37 °C for a period of 1–3 h. The nanomaterial was characterized using UV, EDX, XRD, and HR-TEM techniques, which resulted in spherical and hexagonal morphologies with particle sizes ranging from 3 to 30 nm [115].
(2)$${\mathrm{Ag}}^{+}+{\mathrm{OH}}^{-}\to \mathrm{AgOH}$$
(3)$$2\mathrm{AgOH}\to {\mathrm{Ag}}_{2}O+H_{2}O$$

The biosynthesis of silver nanoparticles has been generously studied over the past decades, with the majority of the nanostructures possessing spherical morphologies and an average diameter of 41 nm (Appendix A). However, numerous researchers encountered silver nanoparticles that were aggregated and polydispersed [116,117,118]. Furthermore, Alghoraibia and co-authors (2019) confirmed that the total phenolic content in the extract solution was directly related to the antioxidant capacity, where a higher concentration of phenolic compounds produced a more substantial reduction in metallic ions [119,120].

### 3.9. Platinum Nanoparticles

The aqueous leaves extract of *Quercus glauca* was reportedly used for the rapid and eco-friendly synthesis of platinum nanoparticles by Karthik and co-authors (2016). The UV-vis spectroscopy of the metallic salt precursor and extract solution displayed peaks at 217 and 273 nm, respectively; these wavelengths diminished with an increase in the absorption value, thereby confirming the complete reduction in Pt (IV) ions to Pt nanostructures. The TEM data provided the morphological details indicating that the platinum nanoparticles had a nearly spherical shape, with particle sizes ranging from 5 to 15 nm [121].

A bio-synthetic green route using *Xanthium strumarium* leaf extract for the synthesis of platinum nanostructures was developed by Kumar and co-authors (2019). The study utilized 2 g of the fresh leaves that were ground to a fine powder mixed with 200 mL deionized water, and boiled for 30 min. The extract solution was then introduced into 190 mL of 1 mM aqueous H_2_PtCl_6_·6H_2_O to complete a volume of 200 mL. The reaction temperature was then increased to 100 °C and left for 1 h. The SEM and TEM images showed that the nanostructures possessed a smooth surface, good alignment and were free from distortion, with cubic to rectangular shapes and an average size of 20 nm [122]. 

Furthermore, Tahir and co-authors (2017) utilized the aqueous extract of the medicinal plant *Taraxacum laevigatum* for the green synthesis of platinum nanoparticles. The nanoparticles were successfully synthesized due to the confirmation sought out by the UV-vis spectroscopy, with the peak originating at 283 nm. The morphological studies displayed well-dispersed spherical nanoparticles with uniform size distribution and an average particle diameter of 2–7 nm (Figure 7a) [123]. Similarly, ultra-small Pt-NPs were produced using hydrothermal [124] and water-in-oil microemulsion [125] techniques that resulted in narrow size distribution and average diameters of 2.45 (Figure 7c) and 5 nm (Figure 7d), respectively. The studies revealed well-defined crystal lattice planes with clusters containing a mixture of irregular and spherical morphologies.

After conducting thorough research through previous studies, it is believed that there has been an attempt to biologically synthesize platinum oxide nanoparticles. Additionally, platinum nanostructures possess the smallest average diameters between the transition-metals examined in this review, with approximately 21 nm (Appendix A). The majority of the nanoparticles retained spherical shapes, however; some studies produced rod-like, wire-like, dodecahedron, hexagonal, and rectangular morphologies [122,126,127,128,129].

### 3.10. Gold Nanoparticles

One of the first reported green synthesis of gold nanoparticles was conducted by Kumar and co-authors (2011), which utilized *Zingiber officinale* extract as a bioreducing and capping agent. The UV-vis spectroscopy confirmed the formation of the gold nanostructures with a surface plasmon resonance (SPR) band emerging at approximately 523 nm. The Dynamic Light Scattering (DLS) study obtained a size distribution curve revealed most particle sizes to be found within the range of 4 to 13 nm [130]. Moreover, aqueous *Cassia auriculata* leaf extract was used in the facile green synthesis of gold nanoparticles [131]. The TEM results displayed spherical, hexagonal, and triangular particle shapes with sizes ranging from 15 to 25 nm. Through the reduction in gold tetrachloride (HAuCl_4_H_2_O), the chemical synthesis of gold nanoparticles was confirmed by the UV-vis spectra with a strong band at 537.2 nm [132]. Interestingly, the study obtained leaf-like morphologies with similar particle diameters to that of the plant-mediated Au-NPs, ranging from 20 to 40 nm.

Dubey and co-authors (2010) conducted their research using the aqueous leaf extract of *Sorbus aucuparia* for the development of gold nanocolloids. An aqueous solution of 10^−3^ M auric acid (HAuCl_4_) was reduced through the addition of 1 mL of the leaf extract at room temperature for 15 min, as seen in Equation (4) [133,134].
(4)$$4H^{+}{\left[{\mathrm{Au}}^{3+}{\mathrm{Cl}}_{4}\right]}^{-}+12H^{+}+12e^{-}\to 4{\mathrm{Au}}^{0}+16\mathrm{HCl}$$

The XRD studies of the substance resulted in peaks at 38.21°, 44.39°, 64.62°, 77.59°, 81.75°, 98.16°, 110.89°, and 115.27°, which is attributed to the face centred cubic structure of gold nanoparticles. Using the Debye–Scherrer equation and the isolated peaks at (111), (311), and (420), the average crystallite size was calculated to be 18 nm.

Once again, the antioxidant capacity of plant extracts was found to be highly dependent on the concentration of phenolic compounds and other phytochemicals, as these biomolecules are considered to be the major contributors to the plants’ reductive capabilities due to their adaptation in response to biotic and abiotic stresses [135]. Additionally, gold nanoparticles were mostly mono-dispersed and possessed spherical morphologies with an average particle size of 45 nm (Appendix A). On the other hand, numerous studies experienced different shapes, including triangular, pentagonal and hexagonal structures [136,137,138]. Furthermore, the clustering and polydispersity of particles were the main obstacles faced by researchers during the biosynthesis of gold nanocrystals [139,140].

## 4. Factors Affecting the Green Synthesis of Nanoparticles

The following section will cover the various parameters that have been previously discovered to affect the characteristics of green synthesized nanoparticles. These include factors such as concentration [141], reaction temperature [142], contact time [143], pH level [144], and calcination temperature [145]. With analyzing how the size and morphologies react to the different parameters, it may be possible to tweak the process to obtain the nanoparticles’ desired characteristics for future researchers.

### 4.1. Concentration

In a study conducted by Anigol and co-authors (2017), silver nanoparticles were synthesized through the use of *Capparis moonii* aqueous extract as a bioreducing and capping agent. The research employed various concentrations of a silver nitrate solution mixed with the extract [146]. More specifically, 250 mL of 1, 2, 3, 4, and 5 mM AgNO_3_ solution were separately mixed with 50 mL of the extract and microwaved for 30 min. As seen in Figure 8d, the nanostructures were characterized by using the UV-vis spectrum, which displayed a surface plasmon absorption band at 430 nm. A general trend was found in which the SPR peak shifted towards the higher wavelength region and became narrower as the concentration increased. A decrease in the nanoparticle diameter accompanied this shift in wavelength; however, this contradicts other studies that have stated a red shift denotes an increase in particle size or agglomeration [139,147,148]. In contrast, the broadening of the absorption band indicates the existence of a more extensive size range within the solution. It was concluded that an increase in the metal salt concentration resulted in the formation of nanostructures with smaller diameters.

Similarly, the aqueous extract of *Citrus paradisi* was utilized in the biosynthesis of silver nanoparticles [149]. In this study, different volumes of the plant extract and silver nitrate (1 mM) solution were used to vary the concentration of the synthesis procedure. This was accomplished by accurately measuring the volumes and mixing the plant extract and metallic salt solutions in ratios of 50:50, 40:60, and 20:80. The reactions were carried out in sunlight for 30 min, where a colour change was then observed. The nanoparticles were disclosed to have spherical morphologies, with a size range of 5 to 65 nm and a mean size of 55.02 nm. The UV-vis analysis of the nanomaterial solution displayed no absorption bands for the 40:60 and 50:50 concentrations; however, the 50:50 volume ratio did reveal a more substantial absorbance than the 40:60 ratio (Figure 8a). On the other hand, the 20:80 volume ratio displayed an intense and broad peak, with the maximum absorption rate being at 420 nm. Faghihi and co-authors (2017) mentioned that the higher absorption indicated the increased formation of silver nanoparticles. Therefore, this suggests that the optimal reaction mixture should mainly consist of a higher metal salt concentration than the extract concentration. On the contrary, Asemani and Anarjan (2019) obtained better results using a lower concentration of metallic precursor, with the optimal ratio having 1 g of ions in 40 mL of the extract solution [150].

The green synthesis of zinc oxide nanoparticles was performed by utilizing *Prunus avium* extract [151]. The effect of concentration of the metal salt solution was studied, where the extract solution was introduced into separate beakers containing zinc nitrate hexahydrate (Zn(NO_3_)·6H_2_O) with concentrations of 0.005, 0.02, 0.05, and 0.3 M. The nanostructures were characterized through UV, SEM, XRD, and FTIR techniques, which confirmed high-purity zinc oxide nanoparticles with hexagonal shapes and an average size of 20.18 nm. Mohammadi and Ghasemi (2018) stated that when the zinc ion concentration was increased, lower quantities of hexagonal structures were produced due to the formation of large anisotropic particles (Figure 8f). This may have been caused by keeping the plant extract volume constant as the phytochemicals are responsible for the capping and stabilization of the nanoparticles. It was concluded that the zinc nitrate concentration of 0.005 M was optimal as there were more functional groups for the zinc ions to interact with [152]. On the other hand, increasing the concentration resulted in larger particles and aggregations due to the competition between the zinc ions and phytochemicals for the nucleation of the nanostructures.

Kumar and co-authors (2017) demonstrated the successful synthesis of silver nanoparticles through the utilization of *Prunus persica* extract. The nanostructures were prepared by adding 10 mL of a 0.01 M AgNO_3_ into five different volumes of plant extract solution, namely, 1.0, 2.0, 3.0, 4.0, and 5.0 mL. The reaction was maintained at room temperature where a colour change was observed after 5 min, indicating the formation of silver nanoparticles. Kumar and co-authors (2017) established that the silver ions above a particular concentration limit might be toxic to the secondary metabolites as they may cause undesired precipitation. However, the precipitation is prevented as the ionic forms are transformed into nanoparticles. As the concentration of the plant extract was increased, the absorbance of the sample generally decreased. This was an indication of an increase in size of the silver compounds, confirmed through various characterization techniques. More specifically, increasing the concentration reduced the size of the particles until, beyond a specific concentration, the particle sizes increased (Figure 8c,e). It was also mentioned that a higher ratio of reducing agent to substrate accelerates the reduction of Ag^+^ to Ag^0^ but diminishes the quality of the nanocomposites [153].

### 4.2. Reaction Temperature

The effect of synthesis temperature was studied during the growth of silver nanoparticles using *Vitex agnus-castus* aqueous extract [155]. The reaction was kept under continuous stirring for various periods at temperatures of 40–80 °C. The nanomaterial formation was monitored by measuring the intensity of the SPR band through the UV-vis spectra, which resulted in the maximum absorption band positioned at 420 nm due to the presence of silver nanostructures. There were apparent differences in the SPR band as the temperature was altered; however, significant differences could only be observed after a reaction period of 2 h. Since the SPR peak is inversely proportional to the diameter of the nanoparticles, it was estimated that the colloids synthesized at 40 °C were smaller than 1 nm; therefore, the particles may not have possessed sufficient stability. Moreover, the UV-vis absorption peaks increased as the temperature increased from 50 to 80 °C, with the fastest formation of the nanoparticles occurring at the highest temperature utilized. A slower increase in the absorption band was observed for the lower temperatures, indicating a delayed particle growth. It was concluded that the rate of nanostructure formation was firmly temperature-dependent, with the optimal temperatures being 60 to 80 °C. This was assumed to be caused by the larger phytochemicals from the biomatter that form the organic coating on the particle's surface which necessitates higher temperatures for growth.

Furthermore, a study conducted by Latif and co-authors (2018) conducted experiments on the green synthesis of gold nanoparticles using *Centella asiatica* extract as a bioreducing and stabilizing agent. The extract solution was introduced into a 0.5 mM gold chloride (HAuCl_4_) aqueous solution in equal volumes, while using process temperatures of 25, 40, 55, and 70 °C under constant stirring. The results displayed a low and broad SPR absorption band for the reaction carried at 25 °C, where it intensified to more sharp and narrow peaks at 542 nm for 40 °C and 536 nm for 55 °C and 70 °C, disclosing an increase in the sphericity of the particles (Figure 9a). Latif and co-authors (2018) deduced that the temperature directly affects the reaction rate as the consumption of gold ions increases, thereby enhancing the formation of nuclei. It was also suggested that the higher temperatures increase the activation energy of the molecules, therefore leading to a faster reaction rate. In addition, the majority of the gold ions are consumed in the nucleation process, hence the high initial reaction rate and thereby, stalling the secondary reduction on the surface of the preformed nuclei, which results in a large population of spherical particles.

In a different experiment, *Cinnamomum camphor* aqueous extract was used in the biosynthesis of silver nanoparticles by Liu and co-authors (2017). The nanostructures were synthesized in two parts with a silver nitrate solution, with sufficient and insufficient Ag^+^ precursors. The researchers evaluated how the size of the nanocolloids were affected by altering the temperature of the mixture from 70 to 90 °C, with increments of 5 °C. Initially, the study determined the kinetics of the process, with k_1_ being the nucleation rate constant and k_2_ being the growth rate constant. It was observed that k_1_ increased slightly when the temperature was raised from 70 to 80 °C (Figure 9c). On the other hand, this constant becomes relatively sensitive and sharply rises when temperatures exceed 80 °C. In addition, the growth rate constant increases linearly as the temperature increases. This should not be the case, as higher temperatures are hypothesized to produce smaller nanoparticles. Liu and co-authors (2017) then stated that the phenomenon was caused by either increasing the temperature above the special value of 80 °C, or whether the Ag^+^ precursor was sufficient or not. Experimentation concluded that with sufficient precursor, the diameter of the nanoparticles would increase almost linearly. However, insufficient precursor resulted in a slight increase in particle diameter up to 80 °C and experienced a sharp decrease in size thereafter, as growth would be restricted due to the effective use of Ag^+^ ions for nucleation, leaving the reaction with a lack of Ag^+^ ions for development [156]. 

Similarly, Jon and co-authors (2019) conducted a study that utilized *Zea mays* extract in the green synthesis of gold nanoparticles. Multiple samples were prepared for the experimentation to be carried out at various temperatures, including 0, 5, 30, 45, 75, and 90 °C. As expected, the surface plasmon resonance band had no peak for 0 °C and 5 °C; therefore, it can be assumed that there was little or no formation of gold nanostructures. On the other hand, the UV-vis spectrum displayed strong peaks that narrowed and intensified as the temperature increased from 30 to 75 °C, which indicated that the size range of gold nanoparticles decreased due to an increase in nucleation [157]. Interestingly, this study displayed similar results to the research conducted by Liu co-authors (2017). The intensity of absorption band significantly decreased and simultaneously broadened, which may have been caused by insufficient ions present in the solution, ultimately indicating an increase in the size range of the nanostructures [158].

### 4.3. Contact Time

The effect of reaction time was evaluated during the green synthesis of silver nanoparticles in a study conducted by Ramli and co-authors (2014). The biosynthesis of the silver nanostructures was executed by adding 10 mL of *Elaeis guineensis* extract into 90 mL of 5 mM AgNO_3_ solution, where the mixture was allowed to react under ambient conditions. The researchers investigated reaction time’s effect by analyzing samples through the UV-vis spectrum every 15 min for 4 h (Figure 10d). The UV-vis analysis of the samples displayed a peak at 435 nm, and the intensity of the band increased as time progressed. Initially, the rate of change in absorbance was minuscule for the first 45 min, where it then increased rapidly up to 120 min. The absorption peaks from 120 to 240 min were identified to be considerably similar; therefore, Ramli and co-authors (2014) concluded that increasing the reaction time will increase the rate of reduction in silver ions until the reaction reaches completion [161].

Behravan and co-authors (2019) accomplished the facile green synthesis of silver nanoparticles through *Berberis vulgaris* leaf and root extract. The nanocompounds were synthesized by adding 3 mM of AgNO_3_ solution into 5 mL of the aqueous plant extract and allowed to react for different time intervals—1, 2, 6, 12, and 24 h. The investigation of the samples was carried out by the UV-vis spectrum between 200 and 800 nm. After one hour, suspended particles had indeed been generated as a broad peak was observed at approximately 440 nm. Interestingly, the UV-vis results from 2 and 6 h displayed decreased absorption peak intensity (Figure 10a). The authors of this research attributed this phenomenon to the reaction having sufficient contact time for the particles to grow, leading to reduced peak intensity. Thereafter, the analysis at 12 and 24 h increased substantially compared to the previous peaks at 1, 2, and 6 h. These two peaks were similar in absorbance, indicating that the reaction had reached equilibrium by 12 h [162].

Furthermore, *Musa paradisiaca* extract was used in a study implemented by Ibrahim (2015) for the biosynthesis of silver nanoparticles. The effect of time on the nanocolloids was studied extensively, with analysis intervals occurring at 3, 5, 10, 15, 20, 25, 30, 45 min, 1, 24, 48, 72, and 96 h (Figure 10b). Visual observations found a colour change of the mixture to yellowish-brown within the first 10 min and to reddish-brown after 1 h, indicating the reduction in Ag^+^ ions. Therefore, Ibrahim (2015) stated that the intensity of the colour was directly proportional to the incubation time of the reaction mixture. The UV-vis spectrophotometer confirmed this, as it displayed a characteristic SPR band for silver nanoparticles at 433 nm that increased over. Initially, the rate of reaction was slow for the first 45 min. Thereafter, the tangible rate was observed until the contact time reached 72 h, where it achieved the most substantial SPR absorption peak and, ultimately, reaction equilibrium. The study concluded that this was due to the increase in silver nanoparticles suspended in the solution [163].

Therefore, it can be said that the contact time generally increases the overall concentration of the nanomaterial until an equilibrium is reached. Thereafter, the nanostructures may form agglomerates as the nucleation rate decreases and the growth of the nanoparticles continues. Lastly, depending on the nanoparticle being synthesized and the biological medium used, the time taken to reach equilibrium varies.

### 4.4. pH Level

In a study conducted by Singh and Srivastava (2015), gold nanoparticles were synthesized using *Amomum subulatum* aqueous extract as a strong reducing and stabilizing agent. The research employed five samples containing 10 mL of 1 mM HAuCl_4_·3H_2_O, where the pH level was altered from 3 to 11 with intervals of 2. The effect of the solution’s pH level was examined through the use of a UV-vis spectrophotometer. The investigation found SPR absorption bands at 572, 560, 544, 548, and 562 nm for the pH values of 3, 5, 7, 9, and 11, respectively (Figure 11d). As previously stated, an increase in wavelength has deemed a redshift, usually accompanied by increased particle size. Therefore, Singh and Srivastava (2015) concluded that the size of the gold nanoparticles decreases as the solution changes from acidic (pH 3) to neutral (pH 7), while the particle size increases when the neutral solution is altered to a more basic medium. Since a larger surface area is desired, a pH value of 7 was considered the optimal parameter value [165].

In addition, the aqueous extract of *Zingiber officinal* was used in the biosynthesis of silver nanoparticles, where the effect of the pH level was evaluated [166]. The reaction’s pH was adjusted by varying NaOH volumes, thereby achieving pH values of 6, 8, 10, 12, and 14, respectively. The XRD results for each pH displayed equivalent peaks located at 2*θ* = 38.2°, 44.4°, 64.6°, 77.5° and 81.5° which are indexed to the (100), (200), (220), (311), and (222), respectively (Figure 11a). On the other hand, the peak intensities increased almost linearly from pH 6 to pH 14. This reflected that a more basic solution increases the size and crystallinity of the Ag nanoparticles. The SEM images further analyzed the compounds, which revealed that the nanostructures were predominantly clustered; however, as the pH increased, the formation of the silver nanoparticles was more pronounced. In summary, the study found that the silver nanoparticles can be tuned by altering the hydrothermal conditions and changing the pH values.

Moreover, the green synthesis of silver nanoparticles was successfully carried out through reduction with *Pistia stratiotes* extract [144]. The pH level of the initial mixture was adjusted from 4 to 10 by the addition of either hydrochloric acid (HCl) or ammonium hydroxide (NH_4_OH). The researchers observed a colour change in the solution from yellow to brown within 5–10 min, assuming the formation of silver nanocolloids; this was confirmed by the analysis of the UV-vis spectrum (Figure 11c). The surface plasmon resonance peaks for pH 4 and pH 6 were found to be situated at approximately 330 nm, while the SPR peaks for pH 8 and pH 10 were observed at around 420 nm. Therefore, the acidic medium demonstrated a blueshift, deeming the particles smaller than in the basic medium. In addition to this, the acidity or basicity of the solution clearly affected the absorption bands’ intensity as the synthesis process favoured a more neutral solution. The TEM images of the nanoparticles confirmed the UV-vis results as the average diameters were revealed to be 16.55 nm and 20.41 nm in an acidic and basic medium, respectively.

Furthermore, Al-Radadi (2019) produced research on the effect of pH levels on the biosynthesis of platinum nanoparticles through the use of *Phoenix dactylifera* as a reducing and stabilizing agent. The pH values were varied to 1.5, 3.5, 5.5, and 7.5 through the addition of either 0.1 M HCl or 0.1 M NaOH. According to the TEM images obtained, the pH of the medium had an effect on the morphology of the nanoparticles. It was found that with an increase in alkalinity, the formation of the nanostructures had increased where bundles of platinum were of heterogeneous shape and sizes, ranging from 2.3 to 22 nm. On the other hand, the mixture with a pH of 5.5 exhibited the production of smaller particles that possessed greater sphericity. Thereafter, Al-Radadi (2019) stated that if smaller nanoparticles are desired, a higher value of pH is required for the efficient capping of the platinum nanomaterial [167].

Therefore, depending on the nanomaterial being synthesized, the pH level has proven to affect the size, morphology, and crystallinity of the nanostructures, with a more neutral pH being favored throughout the majority of the studies researched due to the smaller diameters of the particles, thereby denoting a higher reactivity for their applications.

### 4.5. Calcination Temperature

Calcination temperature refers to the temperature at which the nanoparticles are stripped of their volatile substances. This procedure has been known to affect the characteristics of the nanostructures by refining their size, crystallinity and general stability of the nanomaterial.

In a study conducted by Asemani and Anarjan (2019), copper oxide (CuO) nanoparticles were synthesized using *Juglans regia* leaf extract. The copper oxide nanomaterial was fabricated by mixing the extract solution with copper nitrate, where the annealing temperature was altered from 300–500 °C. The experimentation results displayed a correlation between the calcination temperature and the maximum wavelength and absorption of the solution. It was concluded that nanoparticles with smaller diameters were achieved through higher temperatures, in other words, 400–500 °C. 

Similarly, zinc oxide nanoparticles were fabricated using *Anchusa italic* aqueous extract as a bioreducing and capping agent [145]. The effect of calcination temperature on the zinc oxide nanomaterial was then studied by utilizing furnace temperatures of 100 and 200 °C for 2 h. The XRD analysis of the sample prepared at 100 °C only displayed a few of the diffraction peaks for ZnO when compared to the nanoparticles calcined at 200 °C (Figure 12b). Additionally, the peaks were considerably low in intensity, indicating low crystallinity. Azizi and co-authors (2016) stated that high temperatures are capable of providing sufficient kinetic energy to the process, thus improving the crystal structure of the nanoparticles due to the reorder of the atomic arrangement. The average particle diameter of the nanostructures was calculated using Scherrer’s equation, which found particles sizes to be approximately 10.8 and 16.2 nm for 100 and 200 °C, respectively. It was concluded that by increasing the annealing temperature, the crystal structure and the particle sizes were enhanced.

Furthermore, Mfon and co-authors (2020) demonstrated the effect of annealing temperature on *Ocimum gratissimum*-mediated zinc oxide nanoparticles. The study utilized two different calcination temperatures, namely, 250 and 400 °C, for 3-h periods each. The SEM micrographs revealed that the nanomaterial synthesized at 400 °C were spherical and had a wide range of colloids with clusters around larger particles. The nanostructures biosynthesized at 250 °C also possessed spherical morphologies; on the other hand, the substance had a porous quality compared to the latter [169]. The crystalline size of the particles was calculated to be 29 nm and 14 nm for the calcination temperature of 400 and 250 °C, respectively. Therefore, Mfon and co-authors (2020) concluded that increasing the annealing temperature would, ultimately increase the size of the nanoparticles.

## 5. Drawbacks and Future Perspectives

The characteristics of plant-mediated nanoparticles are still relatively unpredictable compared to chemically synthesized nanoparticles; however, the current knowledge on the subject and ongoing research is narrowing the space between known and unknown. Unfortunately, it is difficult to determine and control the properties of nanostructures by basing experimentation on the assumptions of previous research. Contrarily, a general trend may be utilized for further research in the field. Nevertheless, only macroscopic experimental observations were utilized to produce conclusions in most studies in the literature, with no microscopic insight into the mechanisms of the nanocolloids being taken into consideration. However, once this barrier has been broken, it is highly possible that this sustainable technique may be utilized for the mass production of desired nanoparticles with optimal physical properties.

## 6. Conclusions

In conclusion, the biosynthesis of transition-metal nanoparticles and their applications may be summarized accordingly:Plant extract-mediated nanoparticles have significant advantages as they reduce the complexity of the procedure, inhibit the use of toxic chemicals and eliminate harmful by-products through the modification of chemical synthesis processes.All plants have a reductive capability, with species such as terpenoids, phenols, alkaloids, saponins, and carbohydrates possessing different reducing strengths. This affects the concentration, size, and morphology of the desired nanoparticles.Size and morphology give the nanoparticles their unique properties; however, precisely controlling these characteristics is still troublesome.Agglomeration is deemed a major issue amongst nanoparticles as it may negatively affect their characteristics by reducing the available surface area; however, this may be avoided with the addition of a surfactant.During synthesis, an increase in precursor concentration generally increases particle diameter, concentration, or agglomeration.An increase in synthesis temperature increases the reaction rate, thereby increasing the concentration of nanomaterial and narrowing the particle size range, assuming sufficient precursor.Increasing the reaction time subsequently increases concentration, size, or agglomeration.The reaction solution generally favors a more neutral pH value, although increasing the pH level also increases the crystallinity of the material and may induce clusters to form.The Particle diameter and crystallinity of the nanostructures increases with an increase in calcination temperature.

## Figures and Tables

**Figure 1 materials-14-02700-f001:**
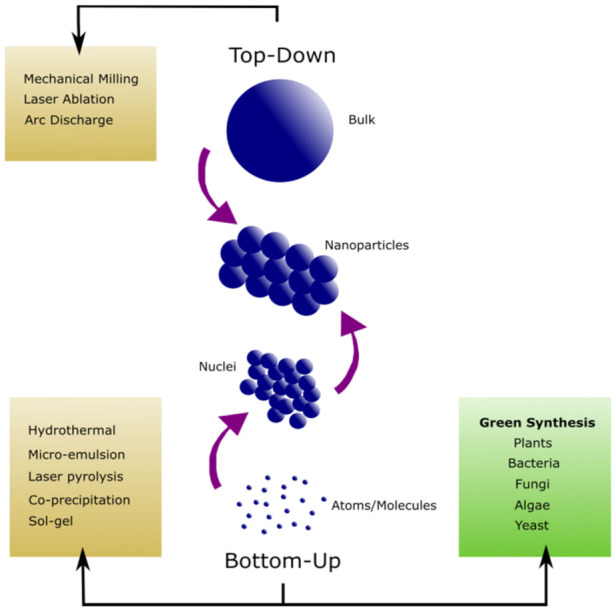
Various Approaches for the Synthesis of Nanoparticles.

**Figure 2 materials-14-02700-f002:**
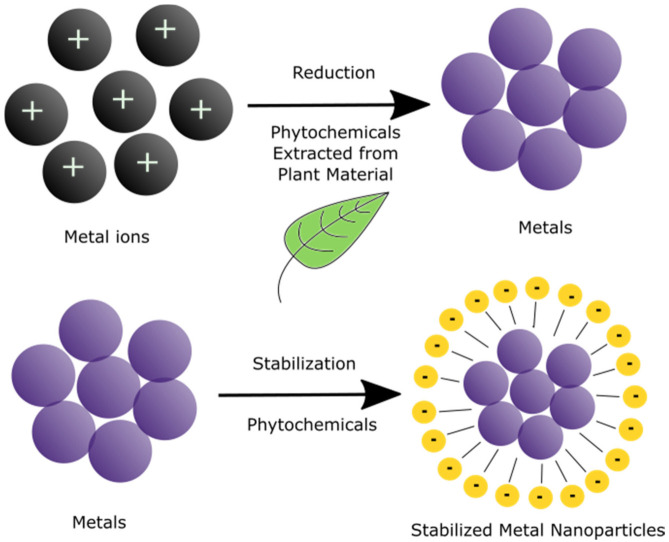
The Growth Mechanism of Nanoparticles through the use of phytochemicals such as Gallic acid, Ferulic acid, Caffeic acid, and Catechin.

**Figure 3 materials-14-02700-f003:**
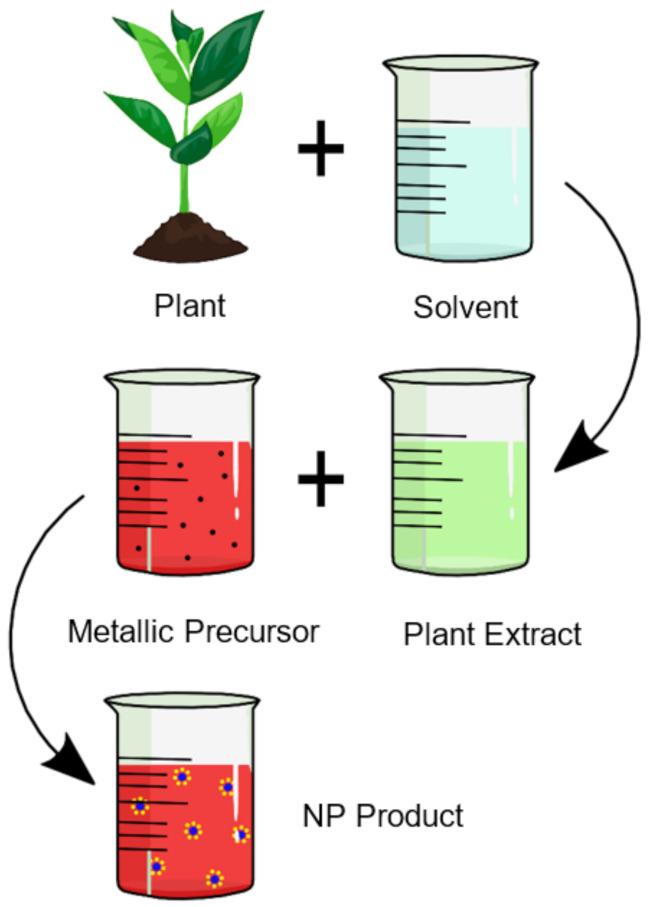
The plant-mediated synthesis of nanoparticles.

**Figure 4 materials-14-02700-f004:**
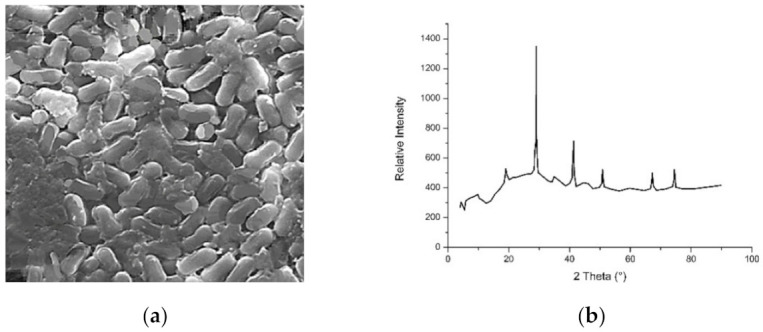
(**a**) The SEM image and (**b**) XRD pattern of Fe_3_O_4_ nanoparticles synthesized using *Caricaya papaya* leaf extract [42]. (with permission from IJSR).

**Figure 5 materials-14-02700-f005:**
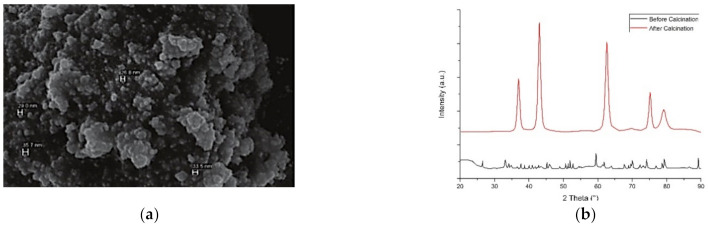
(**a**) SEM image and (**b**) XRD analysis of NiO-NPs synthesized through the use of the sol-gel technique [66]. (This article is available under the Creative Commons CC-BY-NC-ND license and permits non-commercial use of the work as published, without adaptation or alteration provided the work is fully attributed—accessed 04/03/2021).

**Figure 6 materials-14-02700-f006:**
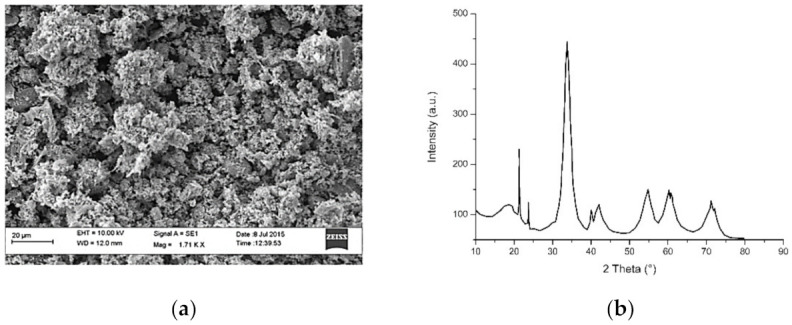
(**a**) SEM image and (**b**) XRD pattern of Pd nanostructures synthesized through the use of *Moringa oleifera* extract [99]. (Permission obtained by Elsevier and Copyright Clearance Center).

**Figure 7 materials-14-02700-f007:**
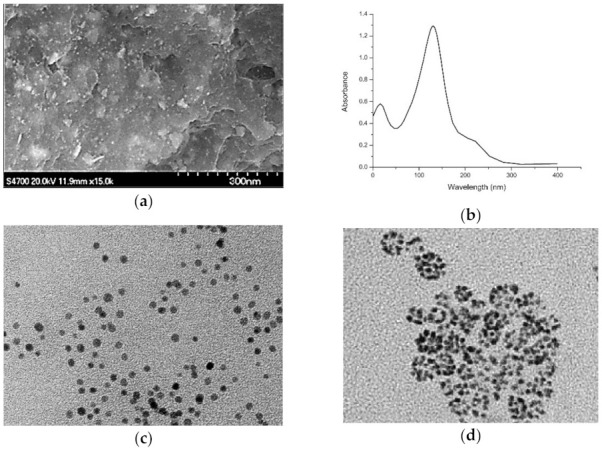
(**a**) SEM image and (**b**) UV-vis spectra of platinum nanoparticles synthesized using *Taraxacum laevigatum* extract [123], and SEM images of Pt-NPs produced through (**c**) hydrothermal [124] and (**d**) microemulsion [125] techniques. (Permission obtained by Elsevier and Copyright Clearance Center for [123,125]. Ref. [124]—This article is an open access article distributed under the terms and conditions of the Creative Commons Attribution license (http://creativecommons.org/licenses/by/4.0/)—accessed 4 March 2021).

**Figure 8 materials-14-02700-f008:**
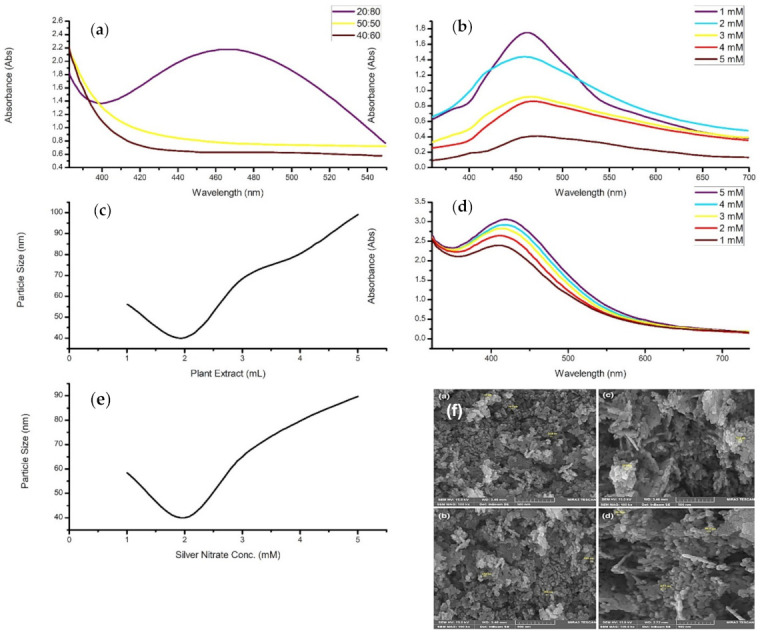
The readapted graphs of (**a**) The effect of plant extract to metal salt ratio on *Citrus paradise*-mediated AgNPs [149]. (**b**) The UV-vis spectra of varying concentrations of AgNO_3_ in the synthesis of AgNPs from *Coleus aromaticus* extract [152]. (**c**) The effect of *Prunus persica* extract volume on AgNP diameter [153]. (**d**) The UV-vis spectrum of different AgNO_3_ concentrations on *Capparis moonii*-conjugated AgNPs [128]. (**e**) The effect of AgNO_3_ concentration on Ag nanoparticle size fabricated through *Prunus persica* [154]. (**f**) SEM of *Prunus avium*-mediated ZnO nanoparticles using different zinc nitrate concentrations [151]. (Ref. [151]—This article is distributed under the terms of the Creative Commons Attribution 4.0 International License (http://creativecommons.org/licenses/by/4.0/), which permits unrestricted use, distribution, and reproduction in any medium, provided you give appropriate credit to the original author(s) and the source, provide a link to the Creative Commons license, and indicate if changes were made—accessed 4 March 2021).

**Figure 9 materials-14-02700-f009:**
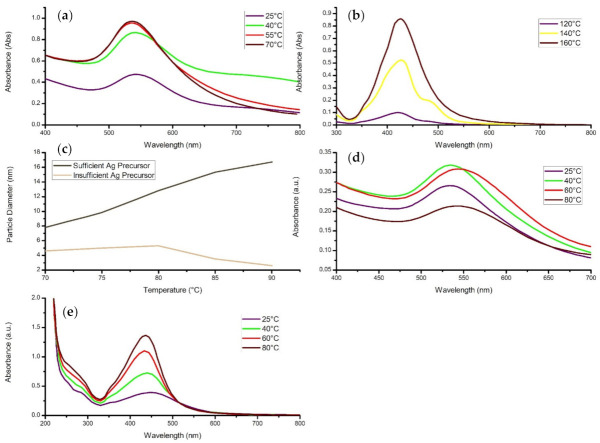
Readapted graphs displaying (**a**) The UV-vis spectrum of *Centella asiatica*-mediated AuNPs at different temperatures [142]. (**b**) The effect of varying synthesis temperatures of green AgNPs [154]. (**c**) The effect of different temperatures with sufficient and insufficient precursor on the size of *Cinnamomum camphor*-mediated AgNPs [156]. (**d**) The UV-vis spectra of AuNPs synthesized using *Chenopodium formosanum* extract through varying the process temperatures [159]. (**e**) The effect of different temperatures on *Eriobotrya japonica*-conjugated AgNPs [160].

**Figure 10 materials-14-02700-f010:**
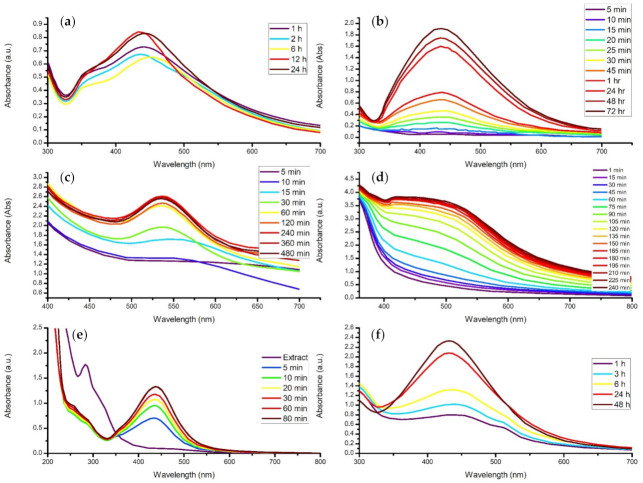
The readapted UV-vis absorption spectrum of (**a**) silver nanoparticles synthesized from *Berberis vulgaris* at different reaction times [162]. (**b**) *Musa paradisa*-conjugated AgNPs at various time intervals [163]. (**c**) The time-dependent synthesis of AuNPs using *Elaeis guineensis* [164]. (**d**) Silver nanoparticles biosynthesized using *Elaeis guineensis* during different time periods [161]. (**e**) *Eriobotrya japonica*-mediated AgNPs at various time intervals [160]. (**f**) Colloidal AgNPs at different reaction times [143].

**Figure 11 materials-14-02700-f011:**
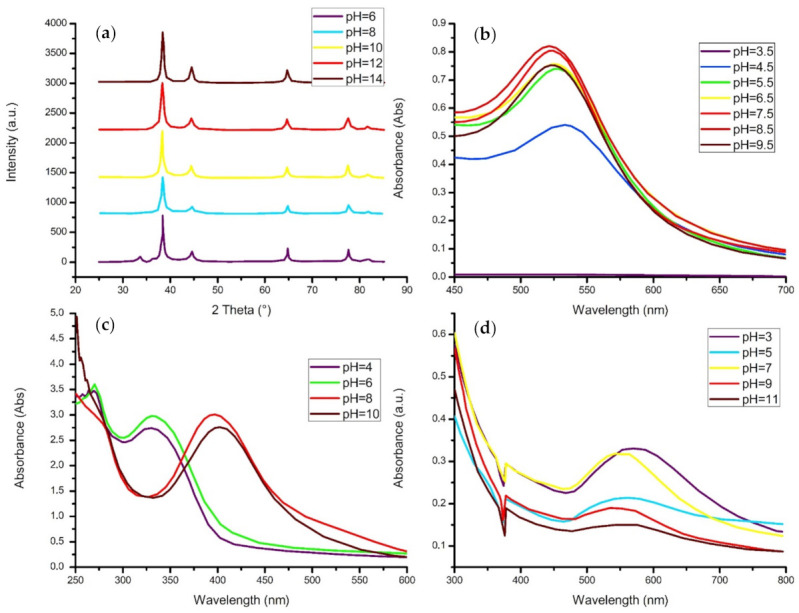
Readapted graphs of the (**a**) XRD pattern of AgNPs synthesized using *Zingiber officinal* under different pH conditions [166]. (**b**) The UV-vis spectrum of the effect of initial pH on *Elaeis guineensis*-mediated AuNPs [168]. (**c**) The UV-vis spectra of AgNPs biosynthesized from *Pistia stratiotes* and how pH levels affect it [144]. (**d**) The UV-vis of *Amomum subulatum*-conjugated AuNPs at different pH values [165].

**Figure 12 materials-14-02700-f012:**
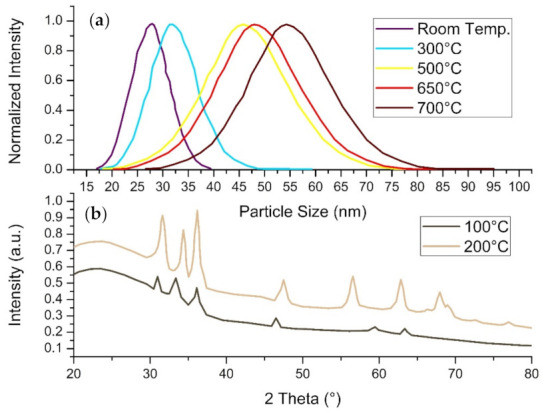
Readapted graphs of (**a**) The UV-vis spectrum of ZnO-NPs and the effect of annealing temperature on particle size [170]. (**b**) The XRD analysis of biosynthesized ZnO nanoparticles using *Anchusa italic* extract [145].

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
