# Peer review of "Green Synthesis of Transition-Metal Nanoparticles and Their Oxides: A Review"

_materials, 2021, doi:10.3390/ma14112700_

Round 1

Reviewer 1 Report

This work "Green Synthesis of Transition Metals Nanoparticle and Their Oxides: A Review" summarizes the green synthesis of transition metal or metal oxide nanoparticles including the methods and obtained materials which is different from the traditional synthesis. This review shows fairly large amount of reference information for the metal/metal oxide synthesis including plant species, size, morphology and its applications, and the factors affecting such green synthesis including concentration, reaction temperature, contact time, pH and calcination temperature. It is comprehensive and detailed, and thus, I suggest this manuscript to be published with a minor revision by Materials Journal after the following comments to be addressed:

Line 102, in the figure caption, it motions to use the plant extracts for preparing nanoparticles, however, we can barely understand it from the image itself to know it is plant extract assisted synthesis, please list it in the figure as well and even consider provides some examples of such organic materials.

Line 122, although the authors show various metals or metal oxides synthesis (from section 3.1 to 3.10) and also supplementary tables, it is better for the authors shown at least one image in the manuscript for the example of certain metal or metal oxide synthesis (e.g. Ti/TiO2) including TEM and/or SEM and/or XRD of the particle size and morphology and its application. With such one example illustration, may help the reader have a direct understanding for a series of examples for different metal/metal oxides synthesis.

Author Response

The response to reviewers comments/suggestion are attached as a file. 

Reviewer 2 Report

Review article by Drummer et al. is focused on the eco-friendly/green synthesis process of the major transition-metal and transition-metal oxide nanoparticles. Plant extract-mediated nanoparticles have been of interest to material scientists as it they inhibit the use of toxic chemicals and eliminate harmful by-products. These methods are also known to increase the complexity of the procedure due to uncontrolled morphologies, poor size distribution and requirements of post synthesis procedures to improve crystallinity.

Authors have covered a wide range of nanoparticles and synthesis procedures in this review. But manuscript lacks the organization at some places. Some of the suggestions are:

  1. Abstract can use some improvement. As for now it is too generalized. Details of the nanoparticles systems discussed in review article are missing from the abstract. Also, a sentence or two on what properties authors are particularly interested in.
  2. Line 43 “Despite this, literature lacks a state-of-the-art review on the application of these compounds” If this is the case why are authors so interested in this specific class of nanomaterials? These materials have wide range of applications spanning from materials to biological sciences. Authors should dig a little deeper in literature.
  3. Line 49 “top-down and bottom-up methods” there are numerous synthesis methods which can be put under there categories. Just like some of them mentioned in the figure1 authors should use some as an example here in text also.
  4. Lines 61 and 62 “problem lies with controlling the size and morphology of the nanoparticles due to the different characteristics and compositions of the biological mediums utilized during synthesis” Authors should highlight any efforts made in literature to improve these factors.
  5. In “2. Biological Synthesis Techniques” Authors have used term “microorganisms” at multiple places. What microorganisms are authors referring to?
  6. Same as in comment 5 which “plants, fungi, bacteria and yeast” are these? Authors need to narrow it down.
  7. In “3. Green Synthesis of Transitional Metals & Their Oxides” Authors should compare non-green routes to green synthesis routes. Factors like morphology, particle size distribution, crystallinity etc. should be discussed for all materials in this section.
  8. Line 473 “parameters that have been previously discovered to affect the characteristics of green synthesized nanoparticles” needs references.
  9. Authors should discuss the importance of aforementioned parameters.
  10. In conclusion Authors have mentioned all the drawbacks but have failed to provide any solution. Again, here authors should make comments on possible solutions to these drawbacks using some literature references.

Author Response

Point by point Responses can be found on the attached document

Reviewer 3 Report

This paper shows a good review of Green Synthesis of Transition Metals Nanoparticle, there are some issues that need to address:

- Introduction is written simply, most recent research and innovation in bionanomaterials performances should be reviewed to show the gap of knowledge. Introduction should be extended with recently research papers.

- resize one of table and transfer from support information to main body of manuscript.

- section of drawbacks and future could be increased quality of manuscript.

- Should be provided a comprehensive part between all of the ionic liquid-based surfactants in the experimental and field scale till now used.

Author Response

Point by point response to the reviewer comments are presented in the attached file.

Reviewer 4 Report

An interesting review article on the uses of nanoparticles, it needs some revisions. My comments are below;

  1. The introduction part is well written, however should discusses industrial scale of green synthesis of such nanoparticles.
  2. Scale up or industrial use, perhaps the possible limitations too.
  3. I suggest to discuss particle shape and sizes (TEM), I suppose figures can be added from other articles by getting copyright permissions. Use the following on how green AgNPs are extracted. 1) Photo–irradiation based biosynthesis of silver nanoparticles by using an ever green shrub and its antibacterial study. 2) Biosynthesis of silver nanoparticles by bamboo leaves extract and their antimicrobial activity.
  4. Variant uses of different aspect ratio of particle size could also be discussed, add similar to these articles for better understandings. 2) Fabrication of alginate fibers loaded with silver nanoparticles biosynthesized via Dolcetto grape leaves (Vitis vinifera cv.): morphological, antimicrobial characterization and in vitro release studies. 3) Solar Irradiation and Nageia Nagi Extract Assisted Rapid Synthesis of Silver Nanoparticles and their Antibacterial Activity.
  5. Conclusion part should be specific, all plants have reducing properties… perhaps what component reduces the nanoparticles should be mentions, phenols, alcohols, so on etc.
  6. It would be nice if industrial use of green synthesized nanoparticles is mentioned, if any.
  7. It has been used in bandages and in some specific use textiles, Mechanically Robust and Antimicrobial Cotton Fibers Loaded with Silver Nanoparticles: Synthesized via Chinese Holly Plant Leaves. Can you comment on their leaching during synthesis or during incorporation to different matrics.

Author Response

Point by point response to the reviewer's comments has been made and attached as a word file.

Round 2

Reviewer 2 Report

Authors made required changes. Manuscript can be accepted in the present form.

Reviewer 3 Report

The article can be accepted for publication.

Reviewer 4 Report

The authors have revised article according to the comments. It can be published.